# Accurate Ship Detection Using Electro-Optical Image-Based Satellite on Enhanced Feature and Land Awareness

**DOI:** 10.3390/s22239491

**Published:** 2022-12-05

**Authors:** Sang-Heon Lee, Hae-Gwang Park, Ki-Hoon Kwon, Byeong-Hak Kim, Min Young Kim, Seung-Hyun Jeong

**Affiliations:** 1School of Electronics Engineering, Kyungpook National University, Daegu 41566, Republic of Korea; 2The Korea Institute of Industrial Technology, Cheonan 31056, Republic of Korea; 3The Oceanlightai. Co., Ltd., Daegu 41260, Republic of Korea; 4Research Center for Neurosurgical Robotic System, Daegu 41566, Republic of Korea; 5School of Mechatronics, Korea University of Technology and Education, Cheonan 31253, Republic of Korea

**Keywords:** convolution neural network, image enhancement, satellite photography, ship detection

## Abstract

This paper proposes an algorithm that improves ship detection accuracy using preprocessing and post-processing. To achieve this, high-resolution electro-optical satellite images with a wide range of shape and texture information were considered. The developed algorithms display the problem of unreliable detection of ships owing to clouds, large waves, weather influences, and shadows from large terrains. False detections in land areas with image information similar to that of ships are observed frequently. Therefore, this study involves three algorithms: global feature enhancement pre-processing (GFEP), multiclass ship detector (MSD), and false detected ship exclusion by sea land segmentation image (FDSESI). First, GFEP enhances the image contrast of high-resolution electro-optical satellite images. Second, the MSD extracts many primary ship candidates. Third, falsely detected ships in the land region are excluded using the mask image that divides the sea and land. A series of experiments was performed using the proposed method on a database of 1984 images. The database includes five ship classes. Therefore, a method focused on improving the accuracy of various ships is proposed. The results show a mean average precision (mAP) improvement from 50.55% to 63.39% compared with other deep learning-based detection algorithms.

## 1. Introduction

Because various ships are produced and their number is increasing worldwide, ship detection technology is important in various areas such as fishing, maritime transport and security, illegal ship identification, maritime zone control and management, and marine and ecosystem conservation. Ship detection is an important research field in remote sensing. Satellite image preprocessing, deep learning-based ship-detection algorithms, and post-processing methods have been presented to improve the detection performance and thereby, increase the accuracy of electro-optical (EO) satellite image-based ship detection.

EO satellite imagery has spectral RGB channels and provides a large database of shape, outline color, and texture. It can be observed at high resolution. However, scenes in the environment (such as clouds, sunlight, shadows from structures, and other adverse conditions) can degrade image quality and negatively affect ship detection performance. In general, unprocessed low-quality images have low recognition and classification performance [1]. Contrast enhancement is used in digital image processing to improve pattern recognition and classification performance in computer vision [2]. It also enhances the visual effect for a clear distinction between objects. This contrast enhancement has resulted in a dramatic increase in the demand for high-quality remote-sensing images. Contrast is an important image quality factor in satellite images. Therefore, contrast enhancement technology is required for a better representation of terrain information and visual recognition of objects in satellite images [3]. Conventional histogram equalization is the most popular algorithm because of its simple and effective implementation. However, the main drawback of histogram equalization is that the result produced is over-enhanced if there are outliers in the histogram [4]. This results in saturation artifacts and a grainy appearance in the resulting enhanced images. In 2012, another contrast enhancement method using adaptive gamma correction with weighting distribution (AGCWD) based on global histogram equalization was proposed [5]. AGCWD is a method based on gamma correction. Furthermore, there are limitations on setting the degree of adaptive image quality and user-adjustable parameters according to the image characteristics. Hence, the proposed method can be considered to have a high degree of freedom. However, significantly bright or significantly dark sensitivities can also cause chroma artifacts and loss of detail. Therefore, the contrast of the satellite image is improved by applying the global feature enhancement pre-processing (GFEP) method. It improves the contrast by omitting outliers in significantly bright or dark areas and resizing the histogram.

The extraction of ship candidates by analyzing satellite images and then detecting ships through a classifier was proposed as a conventional ship detection method [6,7,8]. Subsequently, efficient object-detection algorithms (R-CNN and Fast R-CNN) applying the region proposal technique based on deep neural networks were proposed [9,10]. Faster R-CNN proposes a region proposal network structure and achieves end-to-end training to improve detection efficiency [11]. A single-shot multibox detector (SSD) can access one image to various grids and detect objects of various sizes [12]. However, these object detectors use an axis-aligned bounding box. Due to the fact that the aspect ratio of a ship is largely owing to its characteristics, if the bounding box aligned to the axis is used, the ships attached closely would disappear even when these are predicted correctly by the non-maximum suppression algorithm. In addition, this area contains unnecessary background information [13]. To overcome these shortcomings, ships are detected with an object-detection algorithm using a rotatable bounding box [13,14,15]. The classification result of the predicted object is output as the classification confidence value. The proposed multiclass ship detector (MSD) uses a technique to obtain a ship candidate group by extracting many objects by setting a low confidence score threshold. That is, to obtain as many positive samples as possible from deep learning-based object detectors. The low-reliability threshold is set (through trial and error) to a value wherein the ship detection accuracy is the highest.

Another approach to improving the ship detection performance is post-processing for verifying the predicted ship. The criteria for assessing false positives are established, and ships that satisfied these criteria are selected for verification. In the post-processing process, Zhang (2016) extracted a primary ship candidate group based on the head of the ship shape and ship body outline features by using a deep neural network-based classifier. Furthermore, a methodology for classifying ships using a CNN was presented [16]. In 2017, Yang Liu presented a technique for dividing land and sea areas through edge detection, extraction of ship candidates in the sea area, and detection of ships using CNN-based classification [17]. Among the large number of primary candidates extracted from our proposed MSD, an object with information similar to that of a ship in the land area is falsely detected. Falsely detected objects include containers, highways, forests, small and large buildings, and ports with ship-like information in the land area. A false detection ship excluded by sea-land segmentation image post-processing algorithm is proposed to solve this problem.

The main contributions of this study are summarized below:GFEP: This technique improves contrast by correcting excessive histogram smoothing by reducing the influence of outliers on pixel intensity and improving the visual effect of objects existing in EO satellite images.MSD: While retaining all the advantages of a detector using rotating bounding-box-based Faster R-CNN, a low confidence score threshold is established at the detector end to obtain many primary candidates.FDSESI: A standard is established that can exclude ships that are falsely detected on land using sea and land segmentation images, and robustly exclude false-positive ships based on the established standard.

The remainder of this thesis is structured as follows: Section 2 describes the proposed methods (GFEP, MSD, and FDSESI) in detail. Section 3 provides a detailed description of the experimental method. Section 4 presents the evaluation and results based on the planned experiment. Section 5 describes the characteristics of the proposed method through a discussion of the experiments of this study. Section 6 draws the conclusions.

## 2. Proposal Methods

This section details the proposed ship detection method using EO satellite imagery. Figure 1 illustrates the flow of the proposed methodology.

### 2.1. Global Feature Enhancement Preprocessing

Object information is obscured owing to the shadow effect caused by high terrain or tall buildings. Moreover, the distinction between clustered ships is unclear. An image contrast enhancement technique is applied to enhance pattern recognition and classification performance by improving these visible problems. The contrast enhancement of conventional histogram equalization techniques is vulnerable to outliers in significantly bright and dark regions. Significantly bright or dark terrains in satellite imagery distort the histogram distribution and result in a suboptimal contrast. Among the intensities of the histogram distribution obtained by AGCWD, only those within the range of the object distribution are considered to reduce the influence of outliers. The histogram distribution results of the original image, conventional histogram, and GFEP are presented in Figure 2. It can be observed in Figure 2b that the histogram distribution is expanded excessively. The result of conventional histogram equalization is the absence of optimal contrast between significantly bright and dark areas. Therefore, to reduce the influence of outliers, the histogram is reconstructed by considering only intensities within 3.5 standard deviations of all the intensities. This range is set because significantly bright or significantly dark intensities are pushed toward the edges of the histogram. Therefore, histogram smoothing is performed with values that fall within the lower and upper bounds of ±3.5 standard deviations of the histogram [18]. These enhanced images are used for the training, validation, and testing of deep-learning-based ship detection methods.

### 2.2. Multi-Class Ships Detector

MSD uses the confidence that is the result of the classifier at the end of the deep learning-based object detector. Figure 3 shows the structure of the MSD. A large number of ship candidates are extracted by filtering using a low confidence threshold. That is, it can be applied to any object detector in these structures. The default MSD benchmarked Rbox-CNN [13]. When an axis-aligned bounding box is used, an object is excluded from the NMS algorithm selection if it has a large aspect ratio or the density between the objects is high. However, if a rotated bounding box is used, a bounding box that expresses the object direction can be generated, and objects excluded from candidate group extraction owing to dissatisfaction with the intersection over the union (IoU) threshold in the NMS process can be minimized. In addition, the detection performance is improved by minimizing the background area information unnecessary for object detection, and the MSD accurately predicts the coordinates and angles of random-oriented bounding boxes through stable regression prediction [13]. The angle parameter $\left(t_{\theta }\right)$ is added to the coordinates of the bounding box information to generate a rotated anchor box. The coordinates $\left(t_{x},t_{y}\right)$ represent the center of the rotated bounding box. The height $\left(t_{h}\right)$ and width $\left(t_{w}\right)$ represent the short and long sides, respectively, of the rotated bounding box. To train the RPN, it compares the class label of the predicted object with the ground truth and location of the anchor box to determine the correct prediction. Here, the IoU is calculated using the area of the two anchor boxes and assessed based on the defined threshold to compare the locations. Finally, if the IoU > 0.6 and the predicted object class label is identical to the ground-truth class label, it is classified as a positive sample. If the IoU < 0.3 or the class label is non-identical, it is classified as a negative sample. Samples that are neither positive nor negative do not contribute to the training. Objectness represents whether the anchor box is a background or object through classification. Moreover, the predicted object in the RPN is passed through the diagonal RoI pooling layer (DRoI) and rotated RoI pooling layer (RRoI) to extract features from the proposal region. The RRoI uses an affine transformer to align the rotated bounding box along the horizontal axis to extract features. However, in a ship with a large aspect ratio, the feature information is distorted or disappears owing to the misalignment of the angle. Therefore, distorted features caused by misalignment of the angle occurring in RRoI pooling are alleviated by concatenating the feature information in which the rotated bounding box is diagonally aligned through DRoI pooling [13]. Then, the predicted ship candidate group is obtained by setting it as the low threshold of the confidence score, which is the confidence score of the classifier calculated at the end of the detector. In general, the classifier confidence score threshold can be varied according to the user’s definition. 

Object detection is expressed as a multitask loss (as shown in Equation (4)) to simultaneously minimize classification loss and bounding-box regression loss. Our loss function for the MSD is defined as
(1)$$L_{cls}\left(p,l\right)=-\log pl$$
(2)$$smooth_{L_{1}}\left(x\right)=\{\begin{array}{c} 0.5x^{2},\;if\left|x\right|<1\\ \left|x\right|-0.5,\;otherwise \end{array}$$
(3)$$L_{reg}\left(t^{*},\;t\right)=\underset{i\in \left\{x,\;y,\;w,\;h,\;\theta \right\}}{\sum }smooth_{L_{1}}\left(t_{i}^{*}-t_{i}\right)$$
(4)$$L\left(p,\;l,\;t^{*},t\right)=L_{cls}\left(p,\;l\right)+\lambda \left[l\geq 1\right]L_{reg}\left(t^{*},\;t\right)$$
where l represents the class label of the object, p is the probability distribution of classes calculated by the softmax function, $t_{i}$ is a coordinate vector representing the five parameters (x, y, w, h, θ) that predict the bounding box, and $t_{i}^{*}$ represents the five parameters of the ground truth. The hyperparameter λ controls the balance between the two tasks. When $\left[l\geq 1\right]L_{reg}\left(t^{*},\;t\right)$, if l=0, it is considered a background class, and regression loss is not involved.

### 2.3. False Detection Ship Exclusion by Sea-Land Segmentation Image

Information that distinguishes the sea from other regions in EO satellite images is effective for ship detection [19,20]. As shown in Figure 4, false detection as a ship in the land area may occur owing to the topography of the land, large buildings, roads, and objects similar to ships. Hence, postprocessing is performed with our proposed false detection ship exclusion by sea-land segmentation image (FDSESI) algorithm that excludes objects that are erroneously detected in the land area from among the primary candidates extracted from the MSD. The FDSESI algorithm comprises two steps. First, a deep learning-based semantic segmentation is used to obtain binary images that classify the sea and land. Second, a process of excluding falsely detected objects on land using the location coordinate information of objects detected by MSD is implemented.

#### 2.3.1. Obtain Sea and Land Segment Mask

To segment land and sea regions in EO satellite images clearly, it is necessary to accurately predict the instances in the image in pixel units. Therefore, the semantic segmentation algorithm model is used to apply atrous spatial pyramid pooling for dense feature extraction. The architecture of the semantic segmentation model is shown in Figure 5. The receptive field area is expanded by inserting a pixel value of zero between the kernels as much as the dilation rate, to address various scales [21]. The output stride represents the ratio of the resolution of the input image to that of the output feature map. It maintains the size of the block 4 feature map to reduce information loss. In addition, atrous spatial pyramid pooling is applied to respond effectively to multiple scales [22]. Finally, the pixel unit classification prediction result of the final feature map is output through a full convolution layer. Furthermore, to obtain a size equal to that of the input image, bilinear interpolation is used to update the sampling to the size of the input image, and the image segmentation algorithm has a pixel-level classification. The loss function applied to the segmentation model is cross-entropy [23].

#### 2.3.2. Exclude False-Positive Candidates on Land Region

The detection performance can be improved if erroneously detected objects are excluded by setting the correct sorting criteria in the object-detection task. The problem of false detection by ship-like objects on land (such as docks, cargo, highways, large/small buildings, topography, and forests) with the exclusion of false-positive candidates in land regions by sea and land segment image algorithms is solved. The land and sea areas are classified in pixel units in the image using the sea and land segment images obtained through the process described in Section 2.3.1. The area in which each object exists is determined by mapping the bounding box location of the falsely detected object to the sea and land binary image. To determine whether the mapped object belongs to the land area, the overlap of the predicted object and land area is calculated in units of pixels. Moreover, the pixel values of the land region and sea region in the sea and land segment images are one and 0, respectively. This is shown in Equation (5),
(5)$$\{\begin{array}{c} P\left(h_{i},\;w_{j}\right)=1,\;Land\;Region\\ P\left(h_{i},\;w_{j}\right)=0,\;Sea\;Region \end{array}$$

That is, P(h, w) is a value that classifies land and sea in units of pixels. It can be expressed as a matrix of width and height with a pixel value of zero or one by matching the predicted bounding box to the sea and land binary image. The land and object region IoU per pixel (*LOIoU*) for determining whether the predicted object is located in the land area is shown in Equation (6):(6)$$LOIoU=\frac{\sum_{i=0}^{H}\sum_{j=0}^{W}P\left(h_{i,},w_{j}\right)}{\left(H+1\right)\;*\;\left(W+1\right)}$$

The *LOIoU* is a criterion for determining the region to which the detected object belongs. It can be obtained using the arithmetic mean of this matrix. The higher the *LOIoU* value, the higher the probability that the predicted ship exists in the land region. Conversely, if it is lower, it is considered to be an accurate detection method. Consequently, the predicted object is excluded if the calculated *LOIoU* value is higher than the threshold value (θ). *LOIoU* is calculated for all ground-truth objects, and θ is selected as the maximum value among these. This assessment is given by Equation (7):(7)$$\{\begin{array}{l} False\;Detection,\;LOIoU\geq \;\theta \\ True\;Detection,\;LOIoU<\theta \end{array}$$

The result of the algorithm that excludes false positive candidates on the land region by sea and land segment image is shown in Figure 6.

## 3. Experiments

### 3.1. Dataset for Ship Detection and Segmentation

The collected data are large-scale scene images captured in the Google Earth application. A total of 1984 EO satellite images were obtained at a resolution of 3000 × 3000 pixels. This dataset consisted of RGB images, and sea and land segmentation mask images. The collected images include harbors, shorelines, nearby waters, and vast oceans. These data were collected under various conditions such as shooting time, ship location, and climate variation. Five classes of ships (cargo ships, oil tankers, aircraft carriers, maritime vessels, and warships) are defined. Figure 7 shows class images of the defined ships. 

The total numbers of satellite images in the five classes are 3759, 399, 87, 8419, and 823, respectively. This dataset is named SDS, and its meaning is the same as “Ship Detection and Segmentation”. The EO satellite images and segmentation mask image samples of the SDS dataset are shown in Figure 8. In particular, aircraft carriers have a large hull, and the number of existing aircraft carriers is remarkably small. These hinder the collection of a larger amount of data compared with ships of the other classes. Hence, less data about aircraft carriers were collected compared with the other classes of ships. This causes a class imbalance problem. Therefore, the purpose of our proposed aircraft carrier crop-and-paste in a sea region (ACCP) augmentation technique is to use image samples of aircraft carriers and sea regions to generate new training images, including aircraft carriers. This augmentation technique can solve the class imbalance problem by oversampling images containing fewer frequent objects [24,25]. Annotated aircraft carriers in the training dataset were cut and pasted into the sea region to generate 395 aircraft carrier training samples. The ACCP is illustrated in Figure 9.

Experiments were conducted using the HRSC2016 dataset to verify the generalizability of the proposed method. For the experiment, satellite image datasets that include various ship categories and segmentation annotation mask datasets that segment the sea and land are required. Several satellite image datasets are commonly used by researchers, including DOTA [26], LEVIR-CD [27], and HRSC2016 [28]. DOTA and LEVIR-CD include targets of various categories other than ships. However, these define various ships as one category and do not provide segmentation annotation masks. High-Resolution Ship Collection 2016 (HRSC2016) datasets are collected from Google Earth and six popular harbors. It is provided by Kaggle. The test dataset of HRSC2016 provides various categories of ship samples and segmentation annotation masks. Therefore, experiments are performed using this dataset. A total of 444 EO satellite images and segmentation annotation masks are present in the test sample of HRSC2016.

### 3.2. Training

Prior to learning MSD and Semantic Segmentation, the training, validation, and test datasets are configured in the ratio 8:1:1. A pre-trained ResNet-101 model is used based on the COCO dataset for feature extraction of the training model [29]. The GPU used in the experiment is a TITAN RTX with 24 GB memory. The maximum number of iterations is set as 120,000 during training, the learning rate is provided as a stepwise initial learning rate {0.0005}, step 50 k {0.00005}, and step 100 k {0.000005}. Furthermore, batch size (2), drop out (0.5), and batch normalization is applied. A stochastic gradient descent optimizer synchronized with momentum (0.9) and weight decay (0.0001) is used [30]. The scales, aspect ratios, and angles are set to {64, 128, 192, 256} pixels, {1:3, 1:5, 1:7}, and {$-\frac{\pi }{2},-\frac{\pi }{4},-\frac{\pi }{6},\;0,\;\frac{\pi }{6},\;\frac{\pi }{4},\;\frac{\pi }{2}$}, respectively, to generate anchors. For the deep-learning-based semantic segmentation model for dividing land and sea regions, feature extraction is performed using the pre-trained model ResNet-50 with the COCO dataset [29]. A total of 53.1 k iterations are performed. The learning rate and weight decay are fixed at 0.00005 and 0.0005, respectively. Batch size (2) and dropout (0.5) are applied, and batch normalization is performed. The learning optimization algorithm uses an ADAM optimizer [31].

### 3.3. Experimental Details

First, an experiment is conducted based on the SDS dataset for a comparative analysis between the proposed methodology and other object detection models. In addition, ship detection of Faster R-CNN and SDD using a non-rotating bounding box is performed. Other experimental groups use the rotating bounding box-based detectors R2CNN [14] and Scrdet [15]. Experiments are conducted by applying the proposed GFEP, ACCP, and FDSESI methods to MSD, R2CNN [14], and Scrdet [15]. In the next experiment, the threshold with the highest performance is determined through trial-and-error, i.e., by varying the confidence score threshold of the detection model to which the proposed pre- and post-processing is applied from 0.1 to 0.4 in intervals of 0.1. The results obtained by individually applying GFEP, ACCP, and FDSESI to MSD are analyzed for a performance comparison of the proposed methods. Finally, ablation studies using the GFEP, MSD, and FDSESI methods are performed based on the HRSC2016 dataset.

## 4. Design Evaluation

### 4.1. Evaluation Strategy

For an unbiased evaluation, the learning parameters and backbone are equal to that in Experiment 3.2. In addition, the detection model is evaluated using average precision (*AP*). The evaluation metric of the detection algorithm is to calculate the *AP* by generating precision and recall curves for each of the classes and finally, use the mean average precision (*mAP*) as the average of the classes. The predicted object in the test dataset determines whether the location and class identifications are correct predictions compared with the ground truth. Here, the evaluation of the location is considered a correct prediction when the IoU between the predicted box and the ground truth box exceeds 0.3. If the predicted object satisfies the preceding two conditions, it is counted as a true positive (*TP*). However, it is calculated as a false positive (*FP*) if either condition is not satisfied, and as a false negative (*FN*) if it is defined as the ground truth but not predicted. Therefore, precision is calculated as the proportion of *TP* samples to all the predicted samples, and recall is calculated as the proportion of *TP* samples to the ground truth samples. The formulas for precision and recall are given by Equations (8) and (9), respectively:(8)$$Precision=\frac{TP}{TP+FP}$$
(9)$$Recall=\frac{TP}{TP+FN}$$

To calculate *AP*, recall and precision are calculated on the 2D coordinates of the x- and y-axes. The calculated area for each class is displayed on the coordinates. Similar to the method used in Pascal VOC2010–2012, samples curve at unique recall values (*r*_1_, *r*_2_, …) when the maximum precision reduces [32]. With this variation, the area under the precision-recall curve are measured after the undulations are removed. Then, $p\left(r_{i}\right)$ is sampled, and the *AP* is calculated as the area of the graph, as shown in Equations (10) and (11). In Equation (12), *mAP* is calculated as the average of the *AP* for each class.
(10)$$AP=\sum^{\;}\left(r_{n+1}-r_{n}\right)p_{interp}\left(r_{n+1}\right)$$
(11)$$p_{interp}\left(r_{n+1}\right)=\underset{\tilde{r}\geq r_{n+1}}{\max}p\left(\tilde{r}\right)$$
(12)$$mAP=\frac{1}{N+1}\overset{N}{\underset{i=0}{\sum }}AP_{i}$$

The evaluation metric of segmentation uses the mean intersection over union (*mIoU*) for the segment classes. The *mIoU* is used mainly for the performance evaluation of multiclass segmentation models. In the image predicted using the segmentation model, a class number is stored for each pixel. Using this information, the common area is calculated for each class by matching the class number for each pixel of the ground truth. The *mIoU* calculation method is defined as follows: (13)$$mIOU=\frac{1}{N+1}\overset{N}{\underset{i=0}{\sum }}\frac{p_{ii}}{\sum_{j=0}^{N}p_{ij}-p_{ii}+\sum_{j=0}^{N}p_{ji}}$$
where N is the category, i is the predicted value, j is the ground truth value, and $p_{ij}$ is the prediction of class i as class j.

### 4.2. Evaluation Result

#### 4.2.1. Experimental Results Based on SDS Dataset

Table 1 shows the results of the detection model [13,14,15] obtained using the rotating bounding box, ship detection results obtained by applying the proposed framework (ACCP, GFEP, FDSESI) to [13,14,15]. The ship detection results of the detection models [11,12] using the non-rotating bounding box are shown. Therefore, the results of eight ship-detection models are compared. The result of the semantic segmentation model in segmenting sea and land used in FDSESI is *mIoU* = 97.72. This can sufficiently distinguish between land and sea. The detection results are summarized as follows based on Table 1:A comparison of the results of Scrdet with those obtained by applying our framework to Scrdet reveals that the *AP* of cargo ships, aircraft carriers, and warship classes increase by 1.26, 37.5, and 7.07, respectively. However, the *AP* of the oil tankers and maritime vessels decrease by 12.91 and 2.17, respectively.A comparison of the results of R2cnn with the results obtained by applying our framework to R2cnn reveals that the *AP* of cargo ships, aircraft carriers, and warship classes increase by 0.56, 35.74, and 15.08, respectively. However, the *AP* of the oil tankers and maritime vessels decrease by 9.35 and 1.96, respectively.A comparison of the results of MSD with those obtained by applying our framework to MSD reveals that the *AP* of cargo ships, aircraft carriers, maritime vessels, and warship classes increase by 2.01, 54.55, 0.56, and 19, respectively. However, the *AP* of the oil tanker decreases by 11.9.

The application of the GFEP, ACCP, and FDSESI methodologies does not improve the performance of all the classes. However, it increases the Scrdet, R2cnn, and MSD *mAP* by 6.16, 8.02, and 12.84, respectively. This, in turn, improves the performance of the overall detector. 

In general, the evaluation is performed based on the confidence score, which is the classification result of object detection. The confidence score indicates the result of the multiclass classification. Thus, a higher confidence score threshold indicates a lower number of predicted *TP* and *FP* samples. The purpose of this experiment is to predict the maximum number of ship candidates from the MSD ship detector and to determine a confidence threshold that indicates the maximum feasible ship detection performance by removing the *FP* samples with FDSESI post-processing. This is a trial-and-error experiment. To evaluate the validity of a low confidence threshold, based on the “GFEP + ACCP + MSD + FDSESI” method in Table 1, the detection performance results according to the confidence threshold are shown in Table 2. Both *TP* and *FP* increase as the confidence threshold decreases. That is, although the number of correctly predicted objects increases, the number of incorrectly predicted objects also increases. Therefore, the threshold value is determined as *mAP*, including the relationship between *TP* and *FP*. The confidence threshold is 0.2 with the highest *mAP*. Thus, it is verified that the highest ship detection performance can be obtained when the *FP* sample is removed by FDSESI by specifying a confidence score threshold of 0.2 in the MSD.

Table 3 presents the mAP results of the ablation version of our algorithm on the SDS dataset. All the proposed algorithms improve ship detection performance. The results of ship detection using GFEP and FDSESI on the SDS dataset are shown in Figure 10.

#### 4.2.2. Experimental Results Based on HRSC2016 Dataset

Ablation studies are performed on the proposed methodologies GFEP, MSD, and FDSESI by using the HRSC2016 dataset to evaluate their individual detection performance. Table 4 presents the prediction results obtained by applying the proposed methods based on the HRSC2016 dataset. The result of the semantic segmentation model for segmenting sea and land that is used in FDSESI is *mIoU* = 87.05. The MSD confidence score threshold is 0.2. The performance improves by *mAP* 0.89 compared with the existing detection performance when only the pre-processing GFEP is applied. It improves by *mAP* 0.89 compared with the existing detection performance when only the post-processing FDSESI is applied. Finally, it improves by *mAP* 1.84 compared with the conventional detection performance when both GFEP and FDSESI are applied. Thus, it is verified that the ship detection performance of the proposed methodologies improves through experiments based on the HRSC2016 dataset. The results of the ship detection using GFEP and FDSESI based on the HRSC2016 dataset are shown in Figure 11.

## 5. Discussion

In this section, the major and minor advantages of the proposed approach and the disadvantages and weaknesses of the algorithm are described. It also describes the similarities and differences between existing ship detection methods.

First, the major and minor advantages of the proposed approach are discussed. As a major advantage, according to this study, GFEP, a preprocessing algorithm to improve the contrast of satellite images, proposed to reduce the influence of outliers in very bright or dark areas, which is a disadvantage of existing histogram equalization technology. In addition, the accuracy of ship detection is improved by removing *FP* samples using the FDSESI post-processing algorithm. It excludes falsely detected objects on land that have information similar to that of ships. This ship detection pipeline can be applied to deep-learning-based ship detectors and has remarkable performance. In addition, the MSD proposed based on trial-and-error experiments is set to a low confidence score to extract the maximum number of predicted ship candidates. Furthermore, *FP* samples are excluded through FDSESI post-processing to select the confidence score threshold representing the highest ship detection performance. Ablation studies on the SDS and HRSC2016 datasets demonstrate the effectiveness and advancement of the proposed method. In particular, GFEP and FDSESI are advantageous in terms of usability because these can be applied to deep-learning-based object detection technology as pre- and post-processing methods. In addition, as a minor advantage, the SDS dataset has relatively insufficient data on aircraft carriers. This causes problems with class imbalance. To address this, the aircraft carrier detection performance is improved with the ACCP data augmentation method that crops and pastes the aircraft carrier objects existing in the training dataset to the sea area.

Second, the FDSESI post-processing technique has a drawback. It is affected by the performance of land and sea segmentation models. The correct ship detection sample is likely to be excluded if the segmentation performance is low and the land area is divided incorrectly into sea areas. Therefore, the higher the segmentation performance, the fewer cases to exclude TP, and the more cases to remove *FP*. This improves the accuracy of ship detection. Nevertheless, in this study, the performance was improved by dividing the sea and land areas with an accurate segmentation performance and excluding objects that were falsely detected as ships on the land. The bounding box coordinates also affect the regression prediction results. The FDSESI algorithm determines whether the predicted bounding box belongs to the land area or sea area. Therefore, the coordinates of the predicted bounding box must elaborately predict the ship’s position. Otherwise, an error belonging to the land area may occur. Thus, it is suitable for an object detection model that utilizes a rotating bounding box, which expresses the object position more elaborately than an object detector that utilizes an axis-aligned bounding box. In particular, ships located in ports and coastal areas are close to the land area. Therefore, the proposed post-processing would have a more significant effect on the improvement in ship detection accuracy if the segmentation performance is improved and an object detector using a rotating bounding box is used.

Finally, the similarities and differences between the conventional ship detection algorithm and the proposed algorithm are explained. Similarly, classification and detection are performed by learning the information of ships in satellite images using a deep neural network-based network. This method has higher generalization performance and detection accuracy than detection techniques based on heuristic image processing. In contrast, deep learning-based ship detection technology performs ship detection in the range of the entire image size. However, the proposed algorithm improves the accuracy of ship detection by specifying the land area that can be falsely detected as a ship rather than the entire image area, and by extensively removing falsely detected *FP* samples in the land area. In addition, the accuracy of ship detection is enhanced by preprocessing the satellite image using the GFEP contrast enhancement technique to improve the visual effect.

## 6. Conclusions

This study makes the following contributions to the field of ship detection via remote sensing: A deep learning-based ship detection method was proposed by applying image preprocessing methods to the dataset and post-processing methods to improve the detection performance. The proposed method comprises three main parts. First, the contrast of the EO satellite image is improved using GFEP. Second, numerous primary candidates that increase the number of detection objects are extracted by setting a low confidence threshold at the end of the deep-learning-based object detector. Third, post-processing is adapted to exclude false positive objects within the land area. An experimental evaluation revealed that the performance of the proposed approach was 12.84 *mAP* higher than that of a previously studied rbox-CNN-based detection. In addition, the detection performance was increased by extending this method to other models. This aspect can be applied to any ship detector that has a deep-learning-based detection model structure, and its performance can be improved. In addition, an SDS dataset with land and sea mask images to segment land and sea, EO satellite images including wide-area information of the Earth’s surface, and five types of ships for different purposes were introduced. 

## Figures and Tables

**Figure 1 sensors-22-09491-f001:**
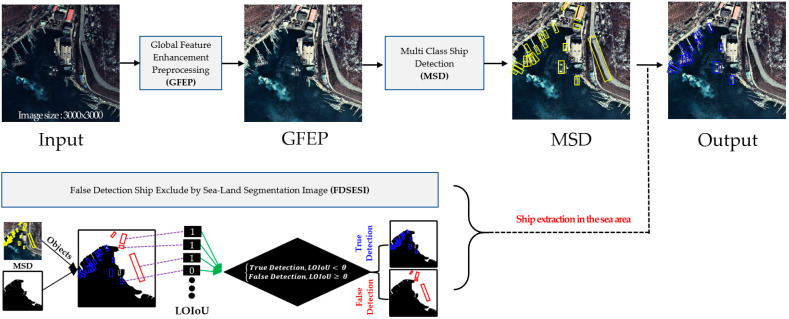
Schematic of proposed methodology.

**Figure 2 sensors-22-09491-f002:**
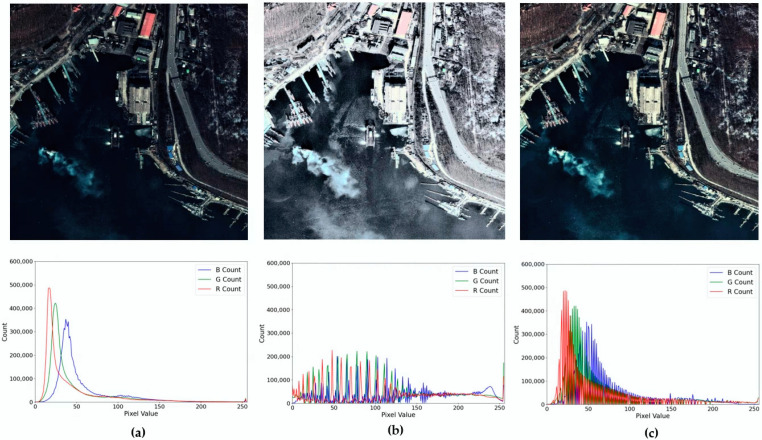
Histogram distribution result according to histogram equalization technique: (**a**) Original image-based histogram; (**b**) Histogram equalization of RGB channels; (**c**) GFEP-based histogram.

**Figure 3 sensors-22-09491-f003:**
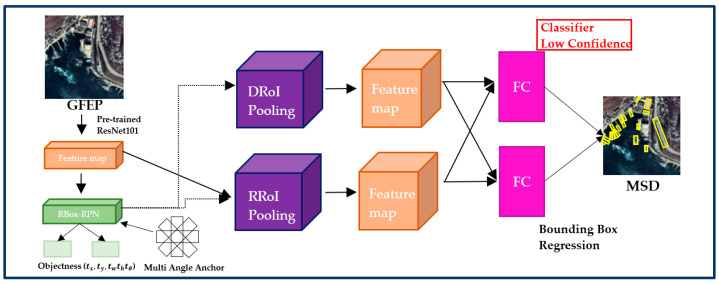
Architecture of MSD for a large number of detection candidates.

**Figure 4 sensors-22-09491-f004:**
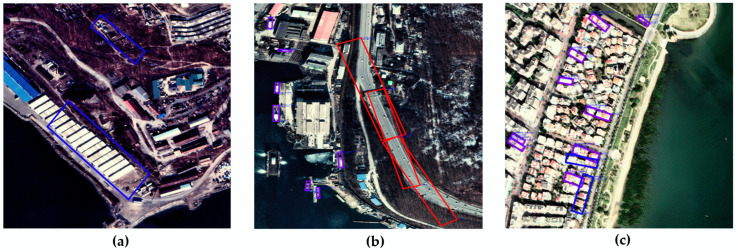
An object falsely detected as a ship within the land area. (**a**) False detection owing to container and topography; (**b**) False detection owing to highway; (**c**) False detection owing to building.

**Figure 5 sensors-22-09491-f005:**
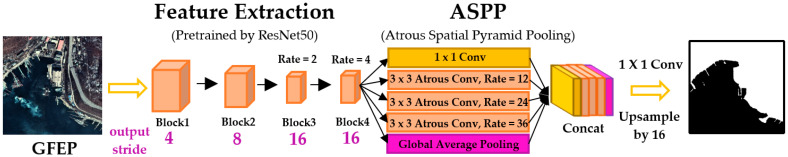
Architecture of semantic segmentation model for classifying land and sea regions.

**Figure 6 sensors-22-09491-f006:**
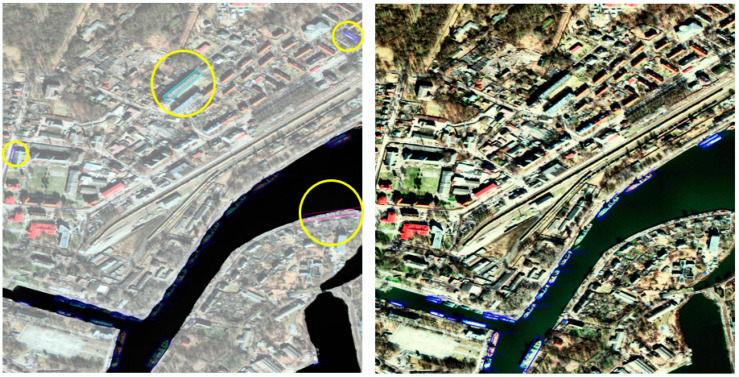
Result of applying FDSESI post-processing. The yellow circle implies false detection on land. The left image implies that the prediction ship of MSD overlaps the area of the sea and land segment mask. The right image implies post-FDSESI.

**Figure 7 sensors-22-09491-f007:**
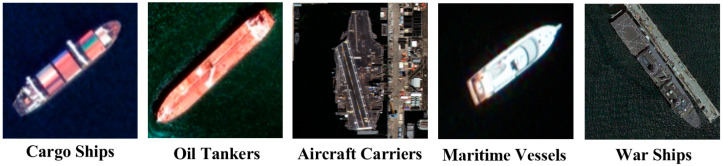
Labeling images for each ship class in the SDS dataset.

**Figure 8 sensors-22-09491-f008:**
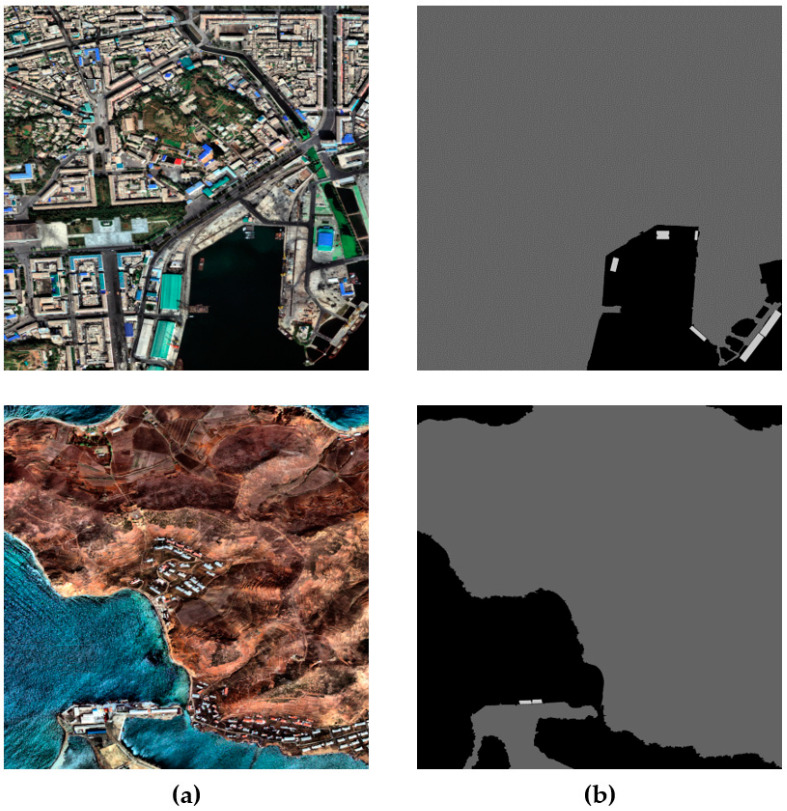
EO satellite image and segmentation mask image samples of the SDS dataset. (**a**) EO satellite image; (**b**) segmentation annotation mask image.

**Figure 9 sensors-22-09491-f009:**
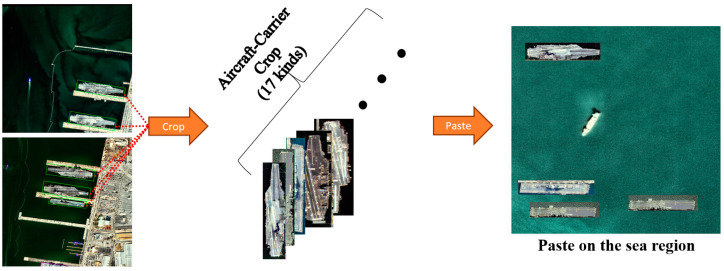
Aircraft carrier data augmentation using crop-and-paste in the sea region.

**Figure 10 sensors-22-09491-f010:**
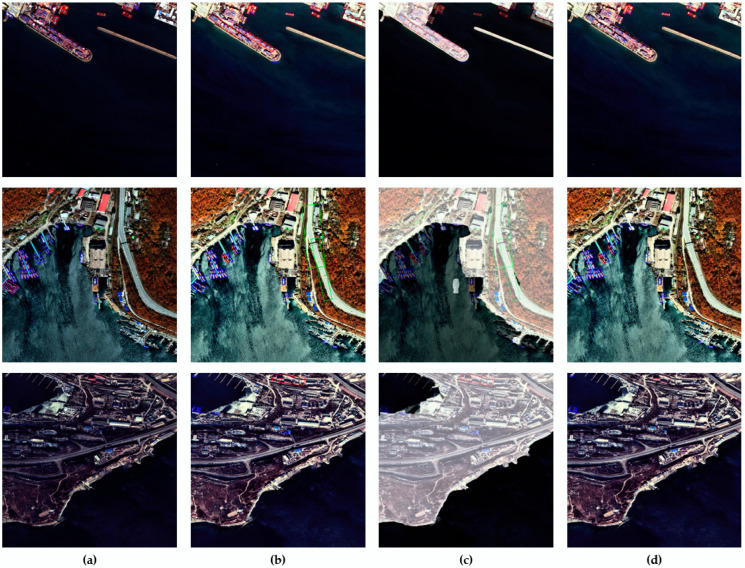
MSD ship detection results applying GFEP and FDSESI based on the SDS dataset. (**a**) Ground truth; (**b**) GFEP; (**c**) Sea, and land segmentation mask overlaid on GFEP; (**d**) Results of applying FDSESI.

**Figure 11 sensors-22-09491-f011:**
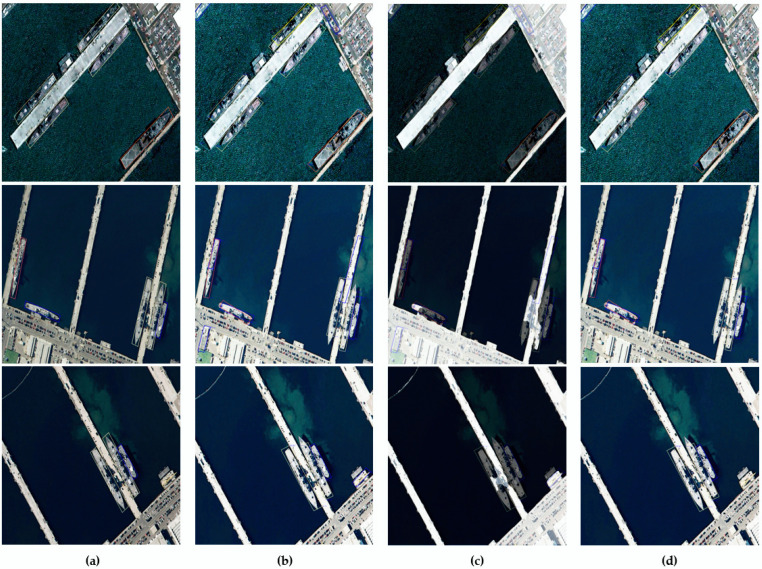
MSD ship detection results applying GFEP and FDSESI based on the HRSC2016 test dataset. (**a**) Ground truth; (**b**) GFEP; (**c**) Sea and land segmentation mask overlaid on GFEP; (**d**) Results of applying FDSESI.

**Table 1 sensors-22-09491-t001:** Results of *AP* by class, *mAP*, and *mIoU* according to the experimental method.

Methods	*AP* (%)						*mIoU* (%)
	car.	oil.	air.	mar.	war.	mean	
Scrdet [15]	81.62	37.25	0	61.84	45.51	45.24	N/A
SSD [12]	70.4	42.59	0	63.25	55.41	46.33	N/A
Faster R-CNN [11]	71.34	45.78	0	60.85	59.48	47.49	N/A
MSD	72.49	**64.21**	0	63.26	52.77	50.55	N/A
R2cnn [14]	82.3	41.34	23.64	61.47	44.36	50.62	N/A
GFEP + ACCP + Scrdet + FDSESI	**82.88**	24.34	37.5	59.67	52.58	51.4	97.72
GFEP + ACCP + R2cnn + FDSESI	82.86	31.99	**59.38**	59.51	59.44	58.64	97.72
GFEP + ACCP + MSD + FDSESI	74.5	52.31	54.55	**63.82**	**71.77**	**63.39**	97.72

N/A means that there is no result of sea and land segmentation used by FDSESI. Bold indicates the highest AP performance for each class.

**Table 2 sensors-22-09491-t002:** Experiment according to variation in confidence threshold to determine the confidence score with the highest performance based on the GFEP + ACCP + MSD + FDSESI method.

Confidence Threshold	0.1	0.2	0.3	0.4
*mAP* (%)	63.36	63.39	58.84	55.56
*TP*	1728	1585	1456	1335
*FP*	748	527	404	319

**Table 3 sensors-22-09491-t003:** Ablation study of proposed methods on the SDS dataset.

Metric	MSD	ACCP + MSD	GFEP + MSD	FDSESI + MSD
*mAP* (%)	50.55	56.60	51.21	58.63

**Table 4 sensors-22-09491-t004:** Ablation study based on HRSC2016 dataset.

MSD	GFEP	FDSESI	mAP
√	-	-	53.33
√	√	-	54.22
√	-	√	54.22
√	√	√	55.17

√ means that the method was applied.

## Data Availability

The datasets generated and/or analyzed during the current study are available from the corresponding author upon reasonable request, subject to the permission of the institutional review boards of the participating institutions.

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
