# Peer review of "Accurate Ship Detection Using Electro-Optical Image-Based Satellite on Enhanced Feature and Land Awareness"

_sensors, 2022, doi:10.3390/s22239491_

Round 1

Reviewer 1 Report

The necessary revisions and comments for authors;

I have re-read and re-evaluated the paper entitled as “Accurate Ship Detection using Electro-optical Image-based Satellite on Enhanced Feature and Land Awareness”. 

There are some major to minor concerns and comments about the submitted article which listed as follows;

1-      This paper has very interesting subject. However, its presentation is not so high quality. Some parts it seems as a report or thesis style. This is a scientific manuscript. Please consider this comment and revise the manuscript for improving the presentation.

2-      Similarly, the manuscript has lack of organization. The figure or flow chart of the method you proposed in the text is explained in introduction section. I think the better understanding, it should be in methodology section.

3-      What is EO satellite image? Most of the commercial satellite images are also used for Earth Observation. Which image you used in the study? What is the specification of the image you used (i.e. spatial resolution, spectral resolution etc.). This information should be given in the text.

4-      In the study, different techniques and proposed algorithm were tested and the average precision (AP) were generated for all. Then the performance comparison were performed using AP. However, I could not see any accuracy assessments. Please prepare some accuracy assessments to show the performances of the methods.

5-      The threshold was applied after performing the proposed approach. Then, the mAP (%) values were obtained. How you decide these threshold values? By trial and error? Or any other technique? If trial and error, why you increase/decrease the threshold level with 0.1 each time? This is because it is also very important for the performance of the proposed approach. Please select more reliable accuracy assessment method for performance evaluation of the proposed technique.

6-      The given satellite images must have scale bar to understand the size of the detected object. Otherwise, they seems as photo not image. Even though, popular application, Google Earth, is used for public use (not so scientific), it has scale bar to identify the object size. Please put scale bars for corresponding images that used in the paper.

7-      There is no any discussion section in the text. If you propose a novel approach, then it is better to give advantageous and disadvantageous of the method. And you must discuss your approach.

8-      It is better not to use so many “we” in the text. For example; we used …, we improved …, we apply …, we set …, we have …, etc. Of course you did it. But this is scientific way of presentation which contains mainly passive tenses.

9-      One of the major concern about the manuscript is the novelty of the paper. What is the innovative part of the study? The proposed approach is a combination of well-known techniques. Therefore, are there any distinct differences between the existing techniques? What are the novel parts or differences from the existing methods? Please write the responses of the questions in the text to explain the novelty for the readers.

Author Response

Thanks for your constructive comments. By following your comments, I revised the manuscript. Please review the answers and corrections below.

Reviewer 2 Report

Comments on “Accurate Ship Detection using Electro-optical Image-based Satellite on Enhanced Feature and Land Awareness”

Please recheck the page limitations of this journal;

Introduction: I suggest the authors point out the shortcomings of the previous studies, however, I couldn’t catch them. The current version is only introducing how the other studies do. The author only gives us the three parts, but readers may want to see why you select to develop GFEP, MSD, and FDI.

Recheck and redraw Figure 1, especially, since the axis of Figure 1b is not clear to me.

Method, how does the GFEP work? The authors present it as an innovation or an important component but lack a clear description that is difficult to accept.

I must say that not all the images can be used to detect ships and improve the accuracy, especially for the high contrast images.

Have you compared the rotating bounding box-based Faster R CNN with the original Faster R CNN or the not rotating bounding box?

The ASPP is popular for the readers. Therefore, I think the author merge some method in this work. What is your innovation in this work?

More detailed information about the EO dataset is needed.

Section 3.1 when did you propose the ACCP augmentation? Not clear.

More comparison for different object detection methods is also needed.

I suggest the author use the public dataset to train first and then test and validate your dataset.

Author Response

(The authors gave the same response as above.)

Round 2

Reviewer 1 Report

After revision, the presentation of the submitted manuscript was improved. However, discussion part, which is newly added by my previous revision note, is not sufficient enough. Discussion must contain major and minor advantages of the proposed approach. In addition, it must contain disadvantageous and weak parts of the algorithm. The similarities and differences of the existing methods should be explained here. My responsibility is to improve the quality of your work. Please re-write the discussion part again according to my comments.

Reviewer 2 Report

I appreciate the authors' effort. More sentences need polish. 
